# Diagnostic accuracy of DPP Fever Panel II Asia tests for tropical fever diagnosis

**Sandhya Dhawan**[1], **Sabine Dittrich**[2,3¤], **Sonia Arafah**[2], **Stefano Ongarello**[2], **Aurelian Mace**[4], **Siribun Panapruksachat**[4], **Latsaniphone Boutthasavong**[4], **Aphaphone Adsamouth**[4], **Soulignasak Thongpaseuth**[4], **Viengmon Davong**[4], **Manivanh Vongsouvath**[4], **Elizabeth A. Ashley**[3,4], **Matthew T. Robinson**[3,4], **Stuart D. Blacksell**[1,3,4]*

**1** Mahidol-Oxford Tropical Research Medicine Unit, Faculty of Tropical Medicine, Mahidol University, Bangkok, Thailand, **2** FIND, Campus Biotech, Geneva, Switzerland, **3** Centre for Tropical Medicine and Global Health, Nuffield Department of Medicine, University of Oxford, Oxford, United Kingdom, **4** Lao-Oxford-Mahosot Hospital-Wellcome Trust Research Unit, Microbiology Laboratory, Mahosot Hospital, Vientiane, Lao PDR

¤ Current address: European-Campus-Rottal-Inn, Deggendorf Institut of Technology, Pfarrkirchen, Germany
* stuart.blacksell@ndm.ox.ac.uk

**Data Availability Statement:** All relevant data are within the manuscript and its Supporting Information files.

## Abstract

### Background

Fever is the most frequent symptom in patients seeking care in South and Southeast Asia. The introduction of rapid diagnostic tests (RDTs) for malaria continues to drive patient management and care. Malaria-negative cases are commonly treated with antibiotics without confirmation of bacteraemia. Conventional laboratory tests for differential diagnosis require skilled staff and appropriate access to healthcare facilities. In addition, introducing single-disease RDTs instead of conventional laboratory tests remains costly. To overcome some of the delivery challenges of multiple separate tests, a multiplexed RDT with the capacity to diagnose a diverse range of tropical fevers would be a cost-effective solution. In this study, a multiplex lateral flow immunoassay (DPP Fever Panel II Assay) that can detect serum immunoglobulin M (IgM) and specific microbial antigens of common fever agents in Asia (*Orientia tsutsugamushi*, *Rickettsia typhi*, *Leptospira spp.*, *Burkholderia pseudomallei*, Dengue virus, Chikungunya virus, and Zika virus), was evaluated.

### Methodology/Principal findings

Whole blood (WB) and serum samples from 300 patients with undefined febrile illness (UFI) recruited in Vientiane, Laos PDR were tested using the DPP Fever Panel II, which consists of an Antibody panel and Antigen panel. To compare reader performance, results were recorded using two DPP readers, DPP Micro Reader (Micro Reader 1) and DPP Micro Reader Next Generation (Micro Reader 2). WB and serum samples were run on the same fever panel and read on both micro readers in order to compare results. ROC analysis and equal variance analysis were performed to inform the diagnostic validity of the test compared against the respective reference standards of each fever agent (S1 Table). Overall better AUC values were observed in whole blood results. No significant difference in AUC

**Funding:** FIND - the Global Alliance for Diagnostics received funding from the Australian and UK governments to provide study coordination and management. This research was funded in whole, or in part, by the Wellcome Trust [220211]. The funders played no role in the study design, data collection and analysis, decision to publish, or preparation of the manuscript.

**Competing interests:** Funding for this work was provided to FIND, the global alliance for diagnostics, by the I have read the journal's policy, and the authors of this manuscript have the following competing interests: FIND, the global alliance for diagnostics, supported Chembio both technically and financially during the development of the DPP 2 assay. MORU provided Chembio with essential reagents necessary for assay development. SDB serves as Section Editor in PLOS Neglected Tropical Diseases Editorial Board.

performance was observed when comparing whole blood and serum sample testing, except for when testing for *R. typhi* IgM (p = 0.04), *Leptospira* IgM (p = 0.02), and Dengue IgG (p = 0.03). Linear regression depicted $R^2$ values had ~70% agreement across WB and serum samples, except when testing for leptosirosis and Zika, where the $R^2$ values were 0.37 and 0.47, respectively. No significant difference was observed between the performance of Micro Reader 1 and Micro Reader 2, except when testing for the following pathogens: Zika IgM, Zika IgG, and *B pseudomallei* CPS Ag.

## Conclusions/Significance

These results demonstrate that the diagnostic accuracy of the DPP Fever Panel II is comparable to that of commonly used RDTs. The optimal cut-off would depend on the use of the test and the desired sensitivity and specificity. Further studies are required to authenticate the use of these cut-offs in other endemic regions. This multiplex RDT offers diagnostic benefits in areas with limited access to healthcare and has the potential to improve field testing capacities. This could improve tropical fever management and reduce the public health burden in endemic low-resource areas.

### Author summary

Tropical fevers, specifically those caused by non-malarial infectious agents, contribute to considerable morbidity and mortality in the Asia-Pacific region. Diagnosis of these pathogens is challenging since the clinical signs are often indistinguishable. Conventional laboratory tests to differentiate between tropical diseases require substantial infrastructure and experienced staff, limiting access to accurate tests in low-resource endemic regions. Rapid diagnostic tools (RDTs) offer an affordable solution for disease management and patient care. Although RDTs are also available for detecting non-malarial pathogens, there are financial and accessibility issues in establishing multiple separate tests in resource-constrained regions. To overcome these challenges, a multi-detection diagnostic platform with the capacity to diagnose a diverse range of tropical fevers would be a solution. This study aimed to evaluate the accuracy of an easier-to-use multiplex lateral flow immunoassay (DPP Fever Panel II Assay) that can detect IgM antibodies and specific antigens of common tropical diseases in Asia (Scrub typhus, Murine typhus, Leptospirosis, Melioidosis, Dengue fever, Chikungunya, and Zika virus). The test performed offers comparable diagnostic accuracy to commercially available tests, as well as some reference tests. The test also performs at equivalent accuracy with both blood and serum samples. If the fever panel were used as a stand-alone test for acute febrile illness diagnosis, cut-offs would need to be adjusted depending on the use of the test, and the desired sensitivity and specificity. There is a need to investigate the use of these cut-offs in other endemic regions, which could improve the rate of tropical fever diagnosis in low-resource settings.

## Introduction

Tropical fever diagnosis has long perplexed healthcare professionals [1,2]. It is well-established that infectious agents are the primary cause of fever-related illness worldwide. In addition to globally prevalent agents, various pathogens are restricted to specific geographical regions and

largely contribute to fever epidemiology in resource-limited settings [3]. In South and Southeast Asia, most of the population lives in rural areas, where poverty rates are high, and healthcare access is limited [4]. Diagnosing and treating diseases in these areas can be challenging due to the limited data available on the causes, resulting in incorrect treatment, including the unnecessary use of antimicrobials. However, it is well documented that febrile illnesses account for substantial morbidity and mortality in these regions [5,6].

While fever is the most frequent and debilitating clinical symptom in the tropics, measures to identify the spectrum of tropical fever aetiology and implement appropriate management measures have been limited [2]. This is especially accurate for non-malarial febrile illnesses. Clinically differentiating between common tropical diseases is challenging because the clinical presentation of fever-causing pathogens is similar. The lack of specific early presentation confounds diagnosis and subsequent treatment [2,7].

The use of rapid diagnostic tests (RDTs) for the early detection of malaria parasites has become common practice over the last decade and aided in improving malaria point-of-care testing globally [8]. As a result, improved case management and control measures significantly decreased the incidence of malarial fever [9], whereas other fever aetiologies proportionally increased [10]. Although single-plex qualitative RDTs for detecting non-malarial fevers are available, there are significant financial and access issues in establishing RDTs for numerous tropical pathogens, both at the patient management and healthcare system level [7]. Once malaria is ruled out, healthcare practitioners are unable to provide further testing and treatment because they receive insufficient training, support, and compensation [2,4,11,12]. As such, curable bacterial infections are often missed during diagnosis [4,13,14], and empiric antibiotic treatments are routinely administered [10,14]. Unnecessary antibiotic use acts as a driver for antimicrobial resistance across communities [15,16]. In low-resource settings where access to laboratory and human resource capacity is constrained, RDTs are preferred for diagnosis because of their affordability and ease of use.

However, RDT kits are designed with set cut-off values that often compromise sensitivity for specificity; in fact, this is a challenge of many serological tests [17]. Thresholds are often selected based on limited samples from one or two regions and often do not take into account varying background seropositivity across different countries, resulting in suboptimal test performance when used outside of the regions tested [7]. There is also a common problem of RDTs of unknown quality being used. While highly sensitive RDTs are vital, tests with low specificity have limited utility in clinical and public health decision-making. Low specificity can lead to high misdiagnosis rates, inappropriate use of antibiotics, and undertreatment of bacterial infections [18–21]. In addition, tests with low specificity can also distort the accuracy of disease estimates, which further hinders the effectiveness of public health response measures [7,18–21].

To overcome some of the delivery challenges of multiple separate tests, a multiplexed RDT with the capacity to diagnose a diverse range of tropical fevers would be a solution. A multiplex assay could deliver significant advantages over current single-plex qualitative RDTs, as they would enable the simultaneous detection and differentiation of numerous infections with comparable clinical manifestations. Additionally, if such a tool is quantitative rather than qualitative as current RDTs, region-specific cut-offs can be used to accomplish defined objectives. Quantitative readings for specific antigens can also serve as indices of severity, as has been shown for histidine-rich protein 2 (HRP2) in malaria [22], capsular polysaccharide (CPS) in melioidosis[23], and non-structural protein 1 (NS1) in dengue [24].

In this study, a multiplex lateral flow immunoassay (DPP Fever Panel II Assay Asia, Chembio, Inc.), that can detect serum immunoglobulin M (IgM) and specific microbial antigens of common fever agents in Asia (*Orientia tsutsugamushi*, *Rickettsia typhi*, *Leptospira spp*.,

*Burkholderia pseudomallei*, Dengue virus, Chikungunya virus, and Zika virus), will be evaluated. The objectives were to assess (i) the diagnostic accuracy of the test in a clinical setting representative of the intended use setting, (ii) compare test performance across whole blood and serum samples, and (iii) assess reader performance variability between two types of micro readers, a DPP Fever Panel II Asia Micro Reader (Micro Reader 1) and the other a DPP Fever Panel II Asia Micro Reader Next Generation (Micro Reader 2).

## Methods

### Ethics statement

The UI-study was approved by the Oxford Tropical Research Ethics Committee (OxTREC, 006–07), and the National Ethics Committee for Health Research in Lao PDR (049/NECHR and 046/NECHR), with approval to use leftover specimens for further research. All patients provided written consent for use of leftover specimens.

### Study population

Specimens were obtained from adult patients (>15 years old) enrolled in the "Prospective study of the causes of fever amongst patients admitted to Mahosot Hospital, Vientiane, Lao PDR" (UI-study) between November 2019 to October 2020. Mahosot Hospital is a main primary-tertiary public hospital in Vientiane (capital of Laos) and receives referrals from across the country. Patients who had fever ($\geq 38°C$) within 24 hours of admission or at enrolment, an illness duration <1 week, a request for blood culture, and leftover paired whole blood and serum volumes of >250μl (following standard diagnostic testing) were enrolled for this study. Samples used for this study were collected from leftover samples on day samples were received, and were stored at 4°C for a maximum of 24 hours prior to testing in this current study.

### DPP fever panel investigation

Whole blood (WB) and serum samples from 300 patients recruited in Vientiane, Laos PDR were tested using the DPP Fever Panel II test, consisting of an Antibody panel and Antigen panel. DPP tests were repeated on samples if they failed. For each patient, testing procedure followed the manufacturer's instructions and were done with both paired blood and serum specimens (to compare specimen suitability), using 50μl of sample for the antigen panel and 10μl of specimen for the antibody panel. To compare reader performance, results were recorded using two DPP readers, DPP Micro Reader (Micro Reader 1) and DPP Micro Reader Next Generation (Micro Reader 2). Whilst diagnostic staff were not blinded to the results of the comparator (S1 Table) and DPP tests, review bias was minimized as the DPP test results do not require interpretation by an operator, only numerical values are displayed by the reader, and the result interpretation was done during data analysis and was not be given to the operator; and pre-specified thresholds for positivity were used for ELISA tests. Targets tested included *O. tsutsugamushi* IgM, *r. typhi* IgM, *Leptospira spp*. IgM, *B. pseudomallei* CPS Ag, Dengue IgM, Dengue IgG, Dengue NS1, Chikungunya IgM, Zika IgM, Zika IgG (Table 1). The test is not yet commercially available; the cutoff values have not been finalised.

### Reference diagnostics

True positives were determined as positives by reference diagnostic tests. The reference diagnostic methods for each pathogen are outlined in S1 Table. For *O. tsutsugamushi* IgM and *R. typhi* IgM detection, an in-house ELISA was used as the reference assay, while for *Leptospira* spp. IgM detection, the SERION ELISA classic Leptospira IgM test was used. The reference

**Table 1. Diagnostic performance of the DPP II Fever Panel Asia on serum versus whole blood.** Summary statistics (cut off, sensitivity, specificity, and AUC values) for the diagnostic performance of whole blood and serum samples run on the panel are depicted. True positives, referring to positives by reference test, have been included as well. Results from both micro readers are shown.

| Pathogen | | | Whole Blood | | | | | | Serum | | | |
|---|---|---|---|---|---|---|---|---|---|---|---|---|
| Micro Reader 1 | Total | True positives | Cut-off | Sensitivity (%) | Specificity (%) | AUC (95% CI) | Total | True positives | Cut-off | Sensitivity (%) | Specificity (%) | AUC (95% CI) |
| *O. tsutsugamushi* IgM | 291 | 21 | ≥4 | 57.1 | 59.6 | 0.61 (0.48–0.74) | 291 | 21 | ≥4 | 42.9 | 55.2 | 0.49 (0.34–0.65) |
| *R. typhi* IgM | 291 | 59 | ≥16 | 76.3 | 73.3 | 0.79 (0.72–0.86) | 291 | 59 | ≥19 | 69.5 | 70.7 | 0.76 (0.69–0.84) |
| *Leptospira spp.* IgM | 291 | 52 | ≥21 | 55.8 | 63.6 | 0.60 (0.51–0.70) | 291 | 52 | ≥19 | 50.0 | 54.0 | 0.53 (0.44–0.62) |
| Dengue IgM | 295 | 36 | ≥7 | 83.3 | 74.1 | 0.85 (0.78–0.92) | 295 | 36 | ≥9 | 75.0 | 76.8 | 0.81 (0.73–0.90) |
| Dengue IgG | 295 | 89 | ≥5.6 | 60.7 | 67.0 | 0.66 (0.60–0.73) | 295 | 89 | ≥6 | 60.7 | 61.7 | 0.64 (0.57–0.71) |
| Chikungunya IgM | 293 | 14 | ≥5.3 | 71.4 | 85.0 | 0.82 (0.67–0.95) | 293 | 14 | ≥6.1 | 78.6 | 88.2 | 0.86 (0.72–0.99) |
| Zika IgM | 291 | 8 | ≥3.6 | 100.0 | 84.5 | 0.97 (0.93–1.00) | 291 | 8 | ≥4.5 | 87.5 | 90.8 | 0.94 (0.89–1.00) |
| Zika IgG | 285 | 66 | ≥2.4 | 59.1 | 57.5 | 0.64 (0.56–0.71) | 285 | 66 | ≥1.9 | 50.0 | 52.1 | 0.53 (0.45–0.60) |
| Dengue NS1 | | | | | | | 294 | 36 | ≥25 | 83.3 | 93.4 | 0.88 (0.80–0.97) |
| *B. pseudomallei* CPS Ag | | | | | | | 283 | 8 | ≥5 | 25.0 | 52.7 | 0.65 (0.13–0.56) |
| **Micro Reader 2** | | | | | | | | | | | | |
| *O. tsutsugamushi* IgM | 291 | 21 | ≥2.9 | 66.7 | 63.0 | 0.71 (0.62–0.79) | 290 | 21 | ≥2.7 | 57.1 | 59.1 | 0.59 (0.45–0.72) |
| *R. typhi* IgM | 291 | 59 | ≥22 | 72.9 | 77.6 | 0.79 (0.72–0.86) | 291 | 59 | ≥22 | 72.9 | 67.5 | 0.75 (0.68–0.83) |
| *Leptospira spp.* IgM | 291 | 52 | ≥21 | 57.7 | 50.2 | 0.59 (0.49–0.69) | 290 | 52 | ≥22 | 51.9 | 52.9 | 0.53 (0.44–0.62) |
| Dengue IgM | 295 | 36 | ≥8 | 77.8 | 78.0 | 0.84 (0.77–0.92) | 295 | 36 | ≥8.5 | 80.6 | 74.5 | 0.84 (0.77–0.91) |
| Dengue IgG | 295 | 89 | ≥3.8 | 70.8 | 54.9 | 0.68 (0.61–0.75) | 295 | 89 | ≥5.4 | 60.7 | 58.3 | 0.64 (0.57–0.71) |
| Chikungunya IgM | 293 | 8 | ≥4.1 | 78.6 | 78.1 | 0.82 (0.70–0.95) | 293 | 14 | ≥4.5 | 71.4 | 72.4 | 0.82 (0.69–0.94) |
| Zika IgM | 291 | 8 | ≥3.7 | 100.0 | 76.7 | 0.94 (0.87–1.00) | 290 | 8 | ≥8.3 | 75.0 | 98.6 | 0.91 (0.79–1.00) |
| Zika IgG | 285 | 66 | ≥2.5 | 50.0 | 49.3 | 0.51 (0.42–0.59) | 284 | 66 | ≥3 | 53.9 | 53.4 | 0.53 (0.44–0.61) |
| Dengue NS1 | | | | | | | 294 | 36 | ≥35 | 86.1 | 93.4 | 0.93 (0.44–0.61) |
| *B. pseudomallei* CPS Ag | | | | | | | 294 | 8 | ≥6 | 75.0 | 72.4 | 0.71 (0.29–0.67) |

test for *B. pseudomallei* CPS detection was blood culture. For both Dengue IgM and NS1 detection, the SD Bioline Dengue Duo IgM/IgG/NS1 (CE Marked) assay was used the reference. While for Chikungunya IgM, Zika IgM, and Dengue IgG, DPP Zika/ Chikungunya/ Dengue multiplex test (CE-marked) was used as the reference assay. Where carried out retrospectively, diagnostic staff were blinded to DPP results.

## Statistical analysis

The data and statistical analysis for this study were performed using Stata/BE 17.0 and R programming language (R 4.1.0). The diagnostic performance of the assays was assessed via sensitivity and specificity. Receiver operating characteristic (ROC) curves were also created using the pROC and ROCR packages on R. The area under the curve (AUC) was examined to inform the diagnostic validity of the test and to advise an appropriate region-specific diagnostic cut-off. A optimal cut-off was selected by maximising both sensitivity and specificity indices from the ROC analysis [25]. A test of equal variance on the AUCs was performed using 'roccomp' and 'rocgold' commands on Stata to inform performance variability between WB and serum samples. Chi-squared hypothesis testing and linear regressions were also conducted to assess the statistical significance of sample type variability and reader variability.

## Results

### Diagnostic accuracy of the DPP Fever Panel in whole blood samples

At an optimal cut-off where sensitivity and specificity are at a suitable compromise, the DPP II *O. tsutsugamushi* IgM test sensitivity was between 57.1–66.7%, while the specificity was approx. 59.6–63.0% (Table 1). *R. typhi* IgM sensitivity was at an appropriate 72.9–76.3%, and specificity was between 73.3–77.6% at an optimal cut-off value. *Leptospira* IgM sensitivity at an optimal cut-off was 55.8–57.7%, and specificity was low at 50.2–63.6% (Table 1). Dengue IgM sensitivity at its optimal range was between 77.8–83.3%, while the specificity was 74.1–78.1%. In comparison, sensitivity for Dengue IgG detection was between 60.7–70.8% and specificity 54.9–67.0% at the optimal cut-off. At an optimal cut-off, Chikungunya IgM detection provided a sensitivity and specificity of 71.4–78.6% and 78.1–85.0%, respectively. Zika IgM detection had a sensitivity of 100%, at 76.7–84.5% specificity. While sensitivity and specificity for Zika IgG detection were compromised, sensitivity was optimal at 50.0–59.0% and specificity at 49.3–57.5% (Table 1).

### Diagnostic accuracy of the DPP Fever Panel in serum samples

At similar cut-off ranges (Table 1), *O. tsutsugamushi* IgM detection in serum samples had an optimal sensitivity of 42.9–57.1% and a specificity of 55.2–59.1%. *R. typhi* IgM sensitivity and specificity were similar, with a sensitivity of 69.5–72.9% and a specificity of 67.5–70.7%. *Leptospira* IgM detection sensitivity at an optimal cut-off was 50.0–51.9%, and specificity was 52.9–54.0%. However, test performance for *B. pseudomallei* CPS Ag displayed greater variance, with sensitivity ranging from 25–75% and specificity ranging from 53–72% (Table 1). Dengue IgM sensitivity and specificity at its optimal range were between 75.0–80.6% and 74.5–76.8%, respectively. While Dengue IgG sensitivity at the optimal cut-off was 60.7%, and specificity was between 58.3–61.7%. Dengue NS1, on the other hand, provided a sensitivity of 83–86% and a specificity of 93.4%. Chikungunya IgM detection had a sensitivity of 71.4–78.6% and a specificity was 72.4–88.2%. At the optimal cut-off, Zika IgM detection was 75.0–78.6% sensitive and 90.8–98.6% specific. While for Zika IgG detection, sensitivity and specificity were compromised, with sensitivity optimal at 50–53.9% with a specificity of 52.1–53.4% (Table 1).

### AUROC analysis

ROC analysis was performed to assess the diagnostic accuracy of the DPP test performance in whole blood samples against serum samples (Fig 1). *O. tsutsugamushi* IgM detection via the DPP Fever Panel II provided an AUC value of 0.61 and 0.71 when run using WB samples and 0.49 and 0.59 with serum samples (Fig 1A). *R. typhi* IgM detection had an AUC of 0.79 across

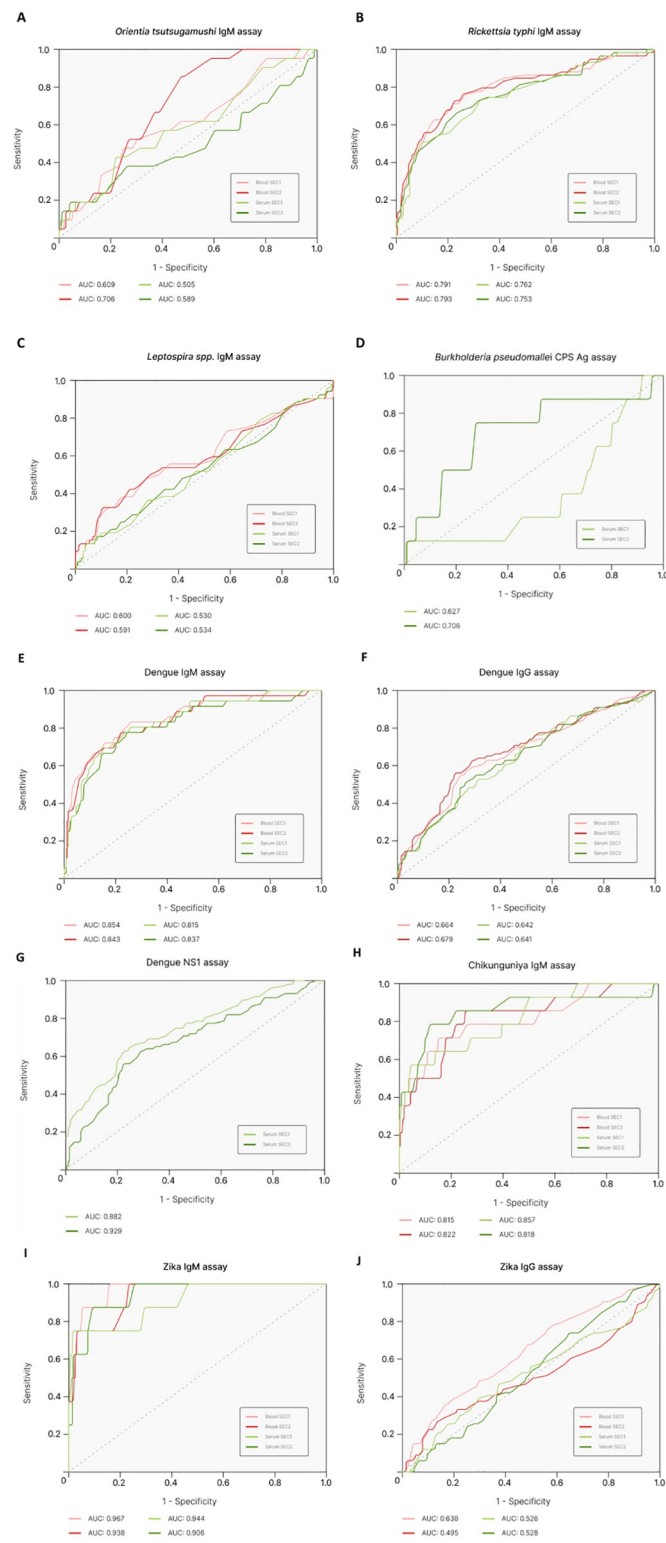

**Fig 1. Receiver Operative Characteristic (ROC) analysis for WB and serum samples.** Area under the curve (AUC) values for WB and serum samples across both readers are shown. No WB samples were available for Dengue NS1 assay and *B pseudomallei* CPS Ag assay. Legend: embedded in the graph.

both readers for WB, with an AUC of 0.76 and 0.75 for serum detection (Fig 1B). The AUC for *Leptospira* IgM detection was 0.60 and 0.59 in WB, while in serum was 0.53 across both readers (Fig 1C). Dengue IgM detection in WB resulted in an AUC of 0.85 and 0.84 and was 0.81 and 0.84 using serum samples (Fig 1E). In comparison, Dengue IgG had an AUC of 0.66 and 0.68 in WB, while IgG detection in serum provided an AUC value of 0.64 (Fig 1F). Chikungunya IgM performed adequately, with an AUC of 0.82 across both readers for WB detection and 0.86 and 0.82 for serum detection (Fig 1H). Zika IgM detection in WB provided an AUC of 0.97 and 0.94 and AUC values of 0.91 and 0.94 in serum detection (Fig 1I). On the other hand, Zika IgG had an AUC value of 0.64 and 0.51 using WB samples and 0.53 when conducted on serum samples. (Fig 1J).

*B. pseudomallei* CPS antigen detection resulted in an AUC of 0.65 and 0.71 on readers 1 and 2, respectively (Fig 1D). The serum samples were also tested for Dengue NS1 detection via the DPP Fever Panel, which provided an AUC value of 0.88 with Micro Reader 1 and 0.93 using Micro Reader 2 (Fig 1G).

## Pairwise comparison of whole blood and serum test performance

Overall better AUC values were observed when WB samples were tested (Table 1). AUC values for whole blood and serum were compared against the gold standard reference results. A test of equality of ROC areas was performed. The AUC variance between whole blood and serum samples ranged from 0.51 to 0.95, with the difference between pathogens being ±0.1 units (Table 2). No significant difference in AUC performance was observed when comparing whole blood and serum sample testing, except for when testing for *R. typhi* IgM (p = 0.04), *Leptospira* IgM (p = 0.02), and Dengue IgG (p = 0.03) (Table 2). The AUC for *R. typhi* IgM WB samples was 0.79, while for serum samples was 0.75. The AUC for *Leptospira* IgM WB samples was 0.59, while for serum samples was 0.53. Dengue IgG WB samples had an AUC value of 0.67, and serum samples had one of 0.64. Linear regression analysis was also conducted to compare WB and serum sample result variance; all outputs were significant. The $R^2$ values generally had ~70% agreement across WB and serum samples, except when testing for Leptospirosis and Zika, where the $R^2$ values were 0.37 and 0.47, respectively (Table 3).

## Pairwise comparison of reader performance

**Whole blood.** Linear regression analysis and ROC test of equal variance were performed to compare performance across both readers. There was no significant difference between reader performances for *O. tsutsugamushi* IgM (*p* = 0.046), *R. typhi* IgM detection (p = 0.872), *Leptospira* IgM (p = 0.317), Dengue IgM (p = 0.466), Dengue IgG (p = 0.209), Chikungunya

**Table 2. Analysis of Equal Variance of WB and serum AUC values.** Pairwise comparison of area under the curve values for whole blood and serum was performed via a chi-square test to deduce variance in performances. Results from both readers (Micro Reader 1 and 2) were compiled to inform robust results.

| Pathogen | Total | AUC (WB) | AUC (Serum) | $\chi^2$ value | p-value |
|---|---|---|---|---|---|
| *O. tsutsugamushi* IgM | 581 | **0.62** | **0.54** | 3.91 | 0.05 |
| *R. typhi* IgM | 581 | **0.79** | **0.75** | 4.31 | 0.04 |
| *Leptospira* spp. IgM | 581 | **0.59** | **0.53** | 5.64 | 0.02 |
| Dengue IgM | 590 | **0.85** | **0.83** | 2.62 | 0.11 |
| Dengue IgG | 590 | **0.67** | **0.64** | 4.76 | 0.03 |
| Chikungunya IgM | 586 | **0.81** | **0.82** | 0.09 | 0.77 |
| Zika IgM | 581 | **0.95** | **0.92** | 1.26 | 0.26 |
| Zika IgG | 569 | **0.58** | **0.51** | 3.75 | 0.05 |

**Table 3. Linear regression analysis of WB and serum diagnostic performance.** WB and serum sample results were directly compared via linear regression to deduce test performance variance across both sample types. Results from both micro readers (1 and 2) were compiled to inform robust results.

| Pathogen | Total | Standard Error | $R_2$ (p-value) | 95% CI |
|---|---|---|---|---|
| *O. tsutsugamushi* IgM | 581 | 0.025 | 0.72 (0.00) | 0.913–1.010 |
| *R. typhi* IgM | 581 | 0.021 | 0.74 (0.00) | 0.796–0.878 |
| *Leptospira* spp. IgM | 581 | 0.035 | 0.37 (0.00) | 0.589–0.727 |
| Dengue IgM | 590 | 0.016 | 0.77 (0.00) | 0.680–0.744 |
| Dengue IgG | 590 | 0.016 | 0.77 (0.00) | 0.680–0.744 |
| Chikungunya IgM | 586 | 0.015 | 0.76 (0.00) | 0.600–0.658 |
| Zika IgM | 581 | 0.024 | 0.47 (0.00) | 0.486–0.579 |
| Zika IgG | 581 | 0.024 | 0.47 (0.00) | 0.486–0.579 |

IgM (p = 0.930), Zika IgM (p = 0.200). There was a significant difference in reader performance for Zika IgG detection (p = 0.004). Although, a linear regression of the reader results suggests similar $R^2$ values for Zika IgM and IgG detection, at 0.662 and 0.664, respectively (Table 4).

**Serum.** There was no significant difference between reader performances for *R. typhi* IgM (p = 0.114), Leptospirosis IgM (p = 0.910), Dengue IgM (p = 0.08), Dengue IgG (p = 0.904), Dengue NS1 (p = 0.124), Chikungunya IgM (p = 0.525), Zika IgM (p = 0.550) and Zika IgG (p = 0.944) (Table 4). There was a 94.1% agreement between reader results ($R^2$, 0.941) for *O. tsutsugamushi* IgM detection; however, a significant difference between reader performances was detected (p = 0.05). *B. pseudomallei* CPS antigen showed no significant difference across reader performance (p = 0.411); though the linear regression revealed an $R^2$ value of 0.0002, it was not a significant output (p = 0.811). While there was no statistical difference between reader performance regarding Zika IgM and IgG detection, the agreement between reader performance was limited. Linear regression analysis displayed an $R^2$ value of 0.435 and 0.439 for Zika IgM and IgG detection, respectively (Table 4).

**Table 4. Linear regression analysis of reader performance.** Reader results from both Micro Reader 1 and Micro Reader 2 were directly compared via linear regression to deduce test performance variance.

| Pathogen | Total | Standard Error | $R_2$ (p-value) | 95% CI |
|---|---|---|---|---|
| **WB** | | | | |
| *O. tsutsugamushi* IgM | 291 | 0.005 | 0.97 (0.00) | 0.767–0.788 |
| *R. typhi* IgM | 291 | 0.009 | 0.96 (0.00) | 0.741–0.776 |
| *Leptospira* spp. IgM | 291 | 0.006 | 0.98 (0.00) | 0.781–0.805 |
| Dengue Ab | 295 | 0.011 | 0.94 (0.00) | 0.756–0.800 |
| Chikungunya IgM | 293 | 0.019 | 0.82 (0.00) | 0.628–0.700 |
| Zika IgM | 291 | 0.026 | 0.66 (0.00) | 0.567–0.669 |
| Zika IgG | 285 | 0.026 | 0.67 (0.00) | 0.569–0.672 |
| **Serum** | | | | |
| *O. tsutsugamushi* IgM | 290 | 0.012 | 0.94 (0.00) | 0.760–0.805 |
| *R. typhi* IgM | 290 | 0.011 | 0.95 (0.00) | 0.738–0.780 |
| *Leptospira* spp. IgM | 290 | 0.008 | 0.97 (0.00) | 0.747–0.779 |
| *B. pseudomallei* CPS Ag* | 281 | 0.062 | 0.00 (0.81) | -0.137–0.107 |
| Dengue NS1* | 291 | 0.011 | 0.93 (0.00) | 0.684–0.728 |
| Dengue Ab | 295 | 0.011 | 0.93 (0.00) | 0.672–0.716 |
| Chikungunya IgM | 293 | 0.010 | 0.95 (0.00) | 0.728–0.766 |
| Zika IgM | 290 | 0.032 | 0.44 (0.00) | 0.410–0.534 |
| Zika IgG | 284 | 0.032 | 0.44 (0.00) | 0.411–0.536 |

## Discussion

This study evaluated the DPP Fever Panel II for the multi-analyte detection of scrub typhus, murine typhus, leptospirosis, melioidosis, dengue fever, chikungunya, and zika virus. The two micro readers (Micro Reader 1 and 2) were screened for performance variability, and the diagnostic platform was assessed using both whole blood and serum samples. Here, test performance was assessed using cutoffs recommended by the manufacturers and region-specific cutoffs calibrated for an optimal level of sensitivity and specificity in endemic settings.

The DPP assay performed poorly when compared to established *O. tsutsugamushi* RDTs, which had greater overall sensitivity (66–84%) and specificity (93–99%) [26–28]. Since it remains unclear how long IgM and IgG antibodies persist in human scrub typhus, samples taken early after symptom onset may not have detectable levels of IgM antibodies [28–30]. Due to the antigenic diversity of *O. tsutsugamushi* strains, cutoffs should be re-evaluated regionally, and local strains included in the antigen pool should be continually updated for accurate clinical diagnosis [31,32].

The DPP assay component for *R. typhi* performed comparably to other RDTs (sensitivity: ~51–60%, specificity: ~94–100%) [33–36], though on the lower end of specificity (67–78%). Little advancements have been made in rapid tests for murine typhus diagnosis [34,37], and it is speculated that the cause of low sensitivity could be the antigenic diversity of *R. typhi* strains geographically, as is the case for *O. tsutsugamushi* [38].

The DPP *Leptospira spp.* IgM assay performed similarly to other RDTs available for leptospirosis diagnosis (sensitivity: 17.9–75%, specificity: 62.1–97.7%) [39–43], albeit at the lower end of specificity. Despite this, the DPP assay obtained consistent sensitivity (~50–58%) and specificity (~50–63%) across sample types, and the diagnostic performance was comparable to earlier used diagnostic tools among healthy slum populations to detect leptospirosis on admission [44]. Commercially available RDTs for the detection of *Leptospira spp.* remain limited in their diagnostic accuracy, none reliably delivering a sensitivity or specificity of >80% on admission [39]. According to published studies, the circulation of location-specific leptospiral serovars contributes to regional variances in background antibody levels [41,45,46], and some serovars may impact the diagnostic accuracy of RDTs [47]. However, the reason region-specific serovars cause more severe illness remains unknown. It is also important to note that anti-*Leptospira* IgM antibodies are not detectable 4–5 days after symptom onset (S2 and S3 Tables) [48,49], and IgM can persist in the blood for years after infection [50,51]. Assays are required to be adjusted to local settings, and samples are collected after a period of seroconversion to avoid false positive results and ensure higher accuracy in diagnosis.

The sensitivity of the DPP *B. pseudomallei* CPS Ag (25%) was comparable to commercially used RDTs for melioidosis (31%) [52], although Micro Reader 2 provided a higher sensitivity (75%) using the regional cutoff. It is well-described that antigen test accuracy in unamplified blood is limited compared to blood culture [53], and only serum samples were tested for *B. pseudomallei* CPS Ag in this study. However, as demonstrated by the DPP test performance, the CPS antigen is not recommended for melioidosis serodiagnosis, as the sensitivity remains lower than culture, the current gold standard (60%) [54,55].

It should be noted that previous studies demonstrate clear associations between CPS positivity and fatality among melioidosis patients [7,23]. By examining the relationship between CPS-positives and disease severity/mortality, we can further investigate the biomarker capacity of the DPP CPS antigen test. The Chembio recommended cutoffs provide a CPS test of higher specificities (95–96%), which could serve high utility in clinical settings to distinguish mild self-limiting illness from severe disease if validated and studied further.

The sensitivity and specificity of the DPP dengue NS1 antigen and IgM antibody were equivalent to that of other RDTs. The Dengue NS1 assay provided greater sensitivity (83–90%) in diagnosis compared to commercially available tests (~45–85%) [56–62]. Commercially available Dengue IgM RDTs provide a diverse range of sensitivity (~20–82%), but generally, studies demonstrate diagnostic sensitivity to be on the lower end of the spectrum [56–59,61]. The DPP Dengue IgM assay specificity is comparable to other IgM RDTs; however, specificity is reduced to ~70% if sensitivity is prioritised. However, the variability in diagnostic accuracy of the DPP Dengue IgM target across WB and serum samples was inconclusive (Table 4).

Further validation studies must be done to confirm the disparities of using whole blood or serum samples. Cutoffs should be adjusted appropriately to represent the region's background seropositivity to achieve desired clinical outcomes. The DPP Dengue IgG assay does not perform as well as the IgM assay and is not comparable to the sensitivity and specificity of readily available Dengue IgG RDTs. There was also a significant difference in assay performance across WB and serum samples. This may be attributed to the average duration of illness, which was observed to be ~6.4 days (S2 and S3 Tables). IgM antibodies are only detectable ~50% of patients 3–5 days after symptom onset [63], and IgG develops latently and may not be detectable for up to 2 weeks after onset of symptoms. A combination of the NS1, IgM, and IgG tests could provide a higher level of accuracy for dengue fever diagnosis. Consistent with previous research, pooling all three analytes, or a combination of two or three, bestowed optimal diagnostic performance (sensitivity ~90%, specificity ~89%) and proved to be of great clinical utility in many low-technology settings [57,61,64–66].

The DPP Chikungunya IgM assay performed at an above-average range, typically in line with the sensitivity (20–100%) and specificity (73–100%) of commercially available RDTs for Chikungunya IgM detection [60,67]. Previous studies document that Chikungunya IgM detection sensitivity increases in the second week after symptom onset [68,69]. The sample collection time is, as such, paramount to ensuring valid test performance and should be considered in the future.

The DPP Zika IgM assay was performed as well as other Chembio Zika IgM RDT assays, showing similar levels of sensitivity (~79–86%) and specificity (~87–100%) [70,71]. The DPP Zika IgG test did not offer high levels of diagnostic accuracy even when cutoffs were optimised to suit regional settings, although IgG detection across RDTs is effective (~90–99%) [71]. IgG levels in Zika infections are often used as a marker of exposure since it develops weeks after onset and can persist in the body for 5–6 months [72,73]. Samples in this study were collected <24h after hospitalisation and may have contributed to lower sensitivity and specificity levels. Further investigation into coinfection rates and cross-reactivity between ZIKV and DENV antigens and antibodies [74,75] is required for diagnosing Zika infections with higher confidence.

The main limitations of the study are the restricted sample size it was conducted in; an absence of true positives (Table 1) can bias sensitivity and specificity, which does not parallel real-life settings. The diagnostic accuracy of the DPP Fever Panel II Asia was overall limited, while the sensitivity of the diagnostic panel is lower than the specificity, it is likely attributed to low levels of antibody during the acute phase of infection [51]. It is recommended to repeat the test after a period of seroconversion to allow for higher confidence in pathogen detection [40]. Further validation should explore cross-reactivity rates as well [76].

Further validation studies are recommended to ensure the synonymous performance of both readers. The performance of both readers was incompatible with one another regarding specific pathogens (Zika IgM, Zika IgG, and *B. pseudomallei* CPS Ag). Further research to explore why WB samples provided better diagnostic accuracy than serum samples overall may

also be of interest. However, they are particularly informative for the company to decide on the most appropriate sample to list in their IFU.

The DPP Fever Panel II Asia offers the opportunity for highly specific rapid multiplex diagnosis of bacterial and arboviral infections. In many low-resource settings, where access to diagnostic infrastructure is limited, introducing an adequately sensitive and specific tool would afford immense benefits for point-of-care clinical management and outbreak surveillance. The DPP Fever Panel II Asia provides quick results without requiring specialised equipment. Given the ease with which the test can be performed, it serves both clinical and field utility, especially when health workers may have limited training [40]. Point-of-care diagnostic tools, particularly biomarker-based and multi-pathogen detection assays, must be prioritised to help guide treatment decisions in decentralised settings [77].

## Supporting information

**S1 Table. Reference diagnostic tests.**
(DOCX)

**S2 Table. Summary statistics for WB assay.**
(DOCX)

**S3 Table. Summary statistics for serum assay.**
(DOCX)

**S1 Fig. Estimated accuracies and 95%-confidence intervals for reader performance.** Error bars are shown. Legend: x-axis, micro readers 1 and 2 (Micro Reader 1 and 2); light pink circles, WB; green circles, serum.
(DOCX)

**S1 Data. Supporting information (raw data).**
(XLSX)

## Acknowledgments

The authors would like to thank the patients and staff of Mahosot Hospital, Vientiane. We thank the staff of the Microbiology Laboratory, Mahosot Hospital, and Dr Susath Vongphachanh Director of Mahosot Hospital and the directors team. We further thank the team from Chembio, particularly Dr Angelo Gunasekera for their technical support and onsite training of staff. We also thank Dr Danoy Chammanam for helping enrol patients.

## Author Contributions

**Conceptualization:** Sandhya Dhawan, Sabine Dittrich, Elizabeth A. Ashley, Matthew T. Robinson, Stuart D. Blacksell.

**Data curation:** Sandhya Dhawan, Sonia Arafah, Aurelian Mace, Matthew T. Robinson, Stuart D. Blacksell.

**Formal analysis:** Sandhya Dhawan, Sonia Arafah.

**Funding acquisition:** Sabine Dittrich, Elizabeth A. Ashley, Matthew T. Robinson, Stuart D. Blacksell.

**Investigation:** Sabine Dittrich, Siribun Panapruksachat, Latsaniphone Boutthasavong, Aphaphone Adsamouth, Soulignasak Thongpaseuth, Viengmon Davong, Manivanh Vongsouvath, Matthew T. Robinson.

**Methodology:** Aurelian Mace, Matthew T. Robinson, Stuart D. Blacksell.

**Project administration:** Sabine Dittrich, Elizabeth A. Ashley, Matthew T. Robinson, Stuart D. Blacksell.

**Resources:** Sabine Dittrich, Elizabeth A. Ashley, Matthew T. Robinson, Stuart D. Blacksell.

**Software:** Sandhya Dhawan, Stefano Ongarello.

**Supervision:** Sabine Dittrich, Elizabeth A. Ashley, Matthew T. Robinson, Stuart D. Blacksell.

**Validation:** Sabine Dittrich, Aurelian Mace, Siribun Panapruksachat, Latsaniphone Boutthasavong, Aphaphone Adsamouth, Soulignasak Thongpaseuth, Viengmon Davong, Manivanh Vongsouvath, Matthew T. Robinson, Stuart D. Blacksell.

**Visualization:** Sandhya Dhawan, Sabine Dittrich, Matthew T. Robinson, Stuart D. Blacksell.

**Writing – original draft:** Sandhya Dhawan, Sabine Dittrich, Matthew T. Robinson, Stuart D. Blacksell.

**Writing – review & editing:** Sandhya Dhawan, Stefano Ongarello, Elizabeth A. Ashley, Matthew T. Robinson, Stuart D. Blacksell.

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
