## [Decision Letter · Decision Letter 0]

30 Jan 2024

Dear Dr. Blacksell,

Thank you very much for submitting your manuscript "Diagnostic accuracy of DPP® Fever Panel II Asia tests for tropical fever diagnosis" for consideration at PLOS Neglected Tropical Diseases. As with all papers reviewed by the journal, your manuscript was reviewed by members of the editorial board and by several independent reviewers. The reviewers appreciated the attention to an important topic. Based on the reviews, we are likely to accept this manuscript for publication, providing that you modify the manuscript according to the review recommendations. 

Please include a paragraph about limitations of the assay in the discussion based on reviewer 1 comments.

Sincerely,

Thomas C Darton

Academic Editor

Shaden Kamhawi

Editor-in-Chief

Reviewer's Responses to Questions

**Key Review Criteria Required for Acceptance?**

**Methods**

-Are the objectives of the study clearly articulated with a clear testable hypothesis stated?

-Is the study design appropriate to address the stated objectives?

-Is the population clearly described and appropriate for the hypothesis being tested?

-Is the sample size sufficient to ensure adequate power to address the hypothesis being tested?

-Were correct statistical analysis used to support conclusions?

-Are there concerns about ethical or regulatory requirements being met?

Reviewer #1: The authors should state how the samples used as "true positives" were confirmed to be so. Was it based on clinical presentation, some other diagnostic test or a combination of both ? When developing and testing a new assay it is crucial that the "known positives" are indeed "genuine true positives".

Reviewer #2: -Are the objectives of the study clearly articulated with a clear testable hypothesis stated? - Yes

-Is the study design appropriate to address the stated objectives? - Yes

-Is the population clearly described and appropriate for the hypothesis being tested? - Yes

-Is the sample size sufficient to ensure adequate power to address the hypothesis being tested? - Authors addressed in limitation

-Were correct statistical analysis used to support conclusions? - Yes

-Are there concerns about ethical or regulatory requirements being met? - Yes

**Results**

-Does the analysis presented match the analysis plan?

-Are the results clearly and completely presented?

-Are the figures (Tables, Images) of sufficient quality for clarity?

Reviewer #1: Yes

What is the "CPS" antigen of B.pseudomallei ? Some readers (including me ) may not know what it is !

Reviewer #2: -Does the analysis presented match the analysis plan? - Suggested for Minor revision 

-Are the results clearly and completely presented? - Yes

-Are the figures (Tables, Images) of sufficient quality for clarity? - Yes

**Conclusions**

-Are the conclusions supported by the data presented?

-Are the limitations of analysis clearly described?

-Do the authors discuss how these data can be helpful to advance our understanding of the topic under study?

-Is public health relevance addressed?

Reviewer #1: This new assay is not all that impressive and I think this needs to be stated in the Conclusions. Its a bit misleading to just say "it is comparable to that of commonly used RDTs", although admittedly these aren't that good either !

Reviewer #2: Suggested for Minor revision

**Editorial and Data Presentation Modifications?**

Reviewer #1: The comparison between the 2 readers is not that interesting to the general reader, although it may well be so to the sponsoring company. This could be summarised in the paper rather than gone into in detail.

"WB" as an abbreviation usually means "Western Blot", not "whole blood". Was it really "whole blood" or was it plasma ? Could another abbreviation be used instead of "WB" ?

Reviewer #2: (No Response)

**Summary and General Comments**

Reviewer #1: Interesting paper but apart from a few assays its not that impressive with only modest sensitivities and specificities for most of the tropical infections tested for. 

line 103. "lack of non-specificity". Don't you mean "lack of specificity" ?

line 144 "will be evaluated" should be "was evaluated"

line 342 "...from severe disease if validated with further study" reads better.

line 360 "igG develops latently" ? Maybe "IgG develops slowly".

line 379 suggest change to "hospitalisation and this early sampling may have contributed to lower sensitivity and specificity for detecting antibody levels."

lines 387-8 suggest change to "Repeating the assay after a period sufficient to allow for seroconversion is recommended to provide greater confidence in the result."

line 414. What is "FIND" ?

line 649. change "compiled" to "combined"

Reviewer #2: The screening and diagnosis of many infectious syndromes using a single sample at the time of a patient visit to a hospital or at the time of a field visit by healthcare workers in the community is very important for timely diagnosis and starting appropriate therapy without delay. The study focuses on this aspect in febrile illness patients with multiplexing the point-of-care tests. I have added a few comments for revision for further comments or recommendations.

PLOS authors have the option to publish the peer review history of their article (what does this mean?). If published, this will include your full peer review and any attached files.

Reviewer #1: No

Reviewer #2: Yes: Sivanantham Krishnamoorthi

Figure Files:

Data Requirements:

Reproducibility:

References

---

## [Editor Report · Decision Letter 1]

18 Mar 2024

Dear Dr. Blacksell,

We are pleased to inform you that your manuscript 'Diagnostic accuracy of DPP® Fever Panel II Asia tests for tropical fever diagnosis Subtitle: Validation of DPP® Fever Panel II' has been provisionally accepted for publication in PLOS Neglected Tropical Diseases.

Best regards,

Thomas C Darton

Academic Editor

Shaden Kamhawi

Editor-in-Chief

Many thanks for addressing the reviewer comments raised, which has been done in a clear and thorough way.

---

## [Editor Report · Acceptance letter]

28 Mar 2024

Dear Dr. Blacksell,

We are delighted to inform you that your manuscript, "Diagnostic accuracy of DPP Fever Panel II Asia tests for tropical fever diagnosis Subtitle: Validation of DPP Fever Panel II," has been formally accepted for publication in PLOS Neglected Tropical Diseases.

Best regards,

Shaden Kamhawi

co-Editor-in-Chief

Paul Brindley

co-Editor-in-Chief
